# More Than a Decade of GeneXpert^®^ *Mycobacterium tuberculosis*/Rifampicin (Ultra) Testing in South Africa: Laboratory Insights from Twenty-Three Million Tests

**DOI:** 10.3390/diagnostics13203253

**Published:** 2023-10-19

**Authors:** Manuel Pedro da Silva, Naseem Cassim, Silence Ndlovu, Puleng Shiela Marokane, Mbuti Radebe, Anne Shapiro, Lesley Erica Scott, Wendy Susan Stevens

**Affiliations:** 1National Health Laboratory Service (NHLS), Johannesburg 2192, South Africa; naseem.cassim@wits.ac.za (N.C.); silence.ndlovu@nhls.ac.za (S.N.); puleng.marokane@nhls.ac.za (P.S.M.); mbuti.radebe@nhls.ac.za (M.R.); wendy.stevens@wits.ac.za (W.S.S.); 2Wits Diagnostics Innovation Hub (DIH), Faculty of Health Sciences, University of Witwatersrand, Johannesburg 2193, South Africa; lesley.scott@wits.ac.za; 3Department of Biostatistics, Boston University School of Public Health, Boston, MA 02118, USA; anshap@bu.edu

**Keywords:** GeneXpert MTB/RIF, GeneXpert MTB/RIF Ultra, molecular diagnostics, tuberculosis, pulmonary tuberculosis, extra-pulmonary tuberculosis, specimen types, implementation

## Abstract

This study seeks to describe the rollout and current state of South Africa’s GeneXpert molecular diagnostic program for tuberculosis (TB). Xpert MTB/RIF was introduced in 2011 with a subsequent expansion to include extra-pulmonary and paediatric testing, followed by Xpert MTB/RIF Ultra in 2017. Through a centralised laboratory information system and the use of a standardised platform for more than a decade, over 23 million tests were analysed, describing the numbers tested, *Mycobacterium tuberculosis* complex detection, rifampin resistance, and the unsuccessful test rates. The stratification by province, specimen type, age, and sex identified significant heterogeneity across the program and highlighted testing gaps for men, low detection yield for paediatric pulmonary TB, and the effects of inadequate specimen quality on the detection rate. The insights gained from these data can aid in the monitoring of interventions in support of the national TB program beyond laboratory operational aspects.

## 1. Introduction

Tuberculosis (TB) is a communicable disease that remains a major cause of morbidity and mortality globally [1]. Geographically, most of the people who developed TB in 2021 were in the World Health Organisation (WHO) regions of Southeast Asia (45%), Africa (23%), and the Western Pacific (18%) [1]. The severity of national TB epidemics, which are reported as the number of incident TB cases, varies widely by country, ranging from less than 5 to more than 500 cases per 100,000 population per year [1]. South Africa ranked as an extremely high TB burden country in 2021, with an incidence of 513 per 100,000 population [1]. The estimated prevalence of bacteriologically confirmed pulmonary TB (PTB) was 852 per 100,000 population among individuals 15 years and older in the 2018 national survey [2]. Furthermore, the HIV co-infection rate among the notified TB cases was 59% [2]. In comparison, the WHO reported that among all the incident cases of TB globally, 8% were people living with HIV [2,3,4]. In addition to the high TB burden and HIV–TB co-infection rates, South Africa also reports high multidrug resistance (MDR-TB) rates [1].

Prior to 2010, the diagnosis of TB disease was based on smear microscopy and/or culture of mycobacterial species [5,6]. A 2011 WHO policy statement indicated a need for novel technologies for rapid TB detection as conventional laboratory testing using smear microscopy was cumbersome and insensitive [6]. A TB-specific, automated, cartridge-based nucleic amplification assay, Xpert^®^ MTB/RIF (Xpert), utilising the GeneXpert (GX) (Cepheid, Sunnyvale, CA, USA) multi-disease platform was developed in response [6]. The Xpert assay detects *Mycobacterium tuberculosis* complex (MTBC) as well as rifampicin (RIF) resistance-conferring mutations directly from a specimen within two hours [6]. The results from a controlled clinical validation reported that the sensitivity of a single Xpert test in smear-negative/culture-positive patients was 72.5%; this increased to 90.2% when three consecutive specimens were tested, with [6] the specificity reported at 99% [6]. Furthermore, Xpert simultaneously detected RIF resistance with 99.1% sensitivity and excluded resistance with 100% specificity [6].

South Africa was the first country to migrate from smear-based to molecular TB diagnostics, implementing Xpert in 2011 [7]. The national implementation was conducted as a phased, forecasted, and managed approach through political endorsement and was supported by both the national treasury and external donors [7].

There have been significant strides made in South Africa to transition TB diagnosis to molecular platforms with improved sensitivity and the ability to provide molecular susceptibility results for RIF, a surrogate marker for MDR-TB, in order to improve treatment outcomes [6,7,8,9].

The objectives of this study were to retrospectively describe the scale-up and maturation of the molecular TB diagnostic program in South Africa over a period of more than a decade since its introduction, with a focus on testing patterns, detection rates, resistance rates, specimen types tested, and demographic trends. These data provide a unique opportunity to describe laboratory perspectives on TB testing trends at the national and provincial levels.

## 2. Materials and Methods

### 2.1. Study Setting

South Africa’s National Health Laboratory Service (NHLS) delivers diagnostic services to the public healthcare system, which includes over 3800 clinics and 400 hospitals [10,11]. The NHLS has a national network of 268 laboratories providing diagnostic pathology services to over 80% of the population [10,11]. Specimens are collected by healthcare workers, with patient and testing details entered manually on an NHLS-specific request form [11,12,13]. These details are then transcribed electronically by data clerks to the national laboratory information system (LIS) upon specimen receipt at the laboratory [11,12]. All the laboratory diagnostic testing platforms are linked to the same national LIS via specific interfaces for direct results transmission and the subsequent reporting [12]. All the LIS data are stored centrally in the NHLS’s data repository [12,14]. Similarly, all the GX platforms are integrated with the LIS [12].

### 2.2. Study Design

The cross-sectional study design was used to analyse the GX laboratory data for the period between 1 March 2011 and 30 April 2023.

### 2.3. Overview of the National GeneXpert Program for TB Molecular Testing

A summary of the national scale-up and implementation is provided below.

#### 2.3.1. Phased Implementation of GeneXpert Platforms

In 2011, the South African National Department of Health implemented the Xpert assay as a replacement test for smear microscopy as the initial diagnostic modality [6]. Following a health economics evaluation which reported that the use of GX technology at the existing decentralised public sector smear microscopy laboratories was substantially less costly than deployment at the points of treatment [8], Xpert testing commenced in March 2011 via a pilot program that placed GX platforms at 25 of the existing smear microscopy laboratories. This was followed by the capacitation of high-burden districts and the rapid scale-up to all smear microscopy laboratories by 2013 [6,9].

The 2011 WHO recommendations for Xpert testing were confined to specimens of pulmonary origin in the adult population, and testing gaps remained [6]. Although Xpert use for TB detection in extra-pulmonary (EPTB) specimens is considered an off-label application, sufficient performance data were gathered [15]; these data guided the updated WHO recommendations for EPTB (all ages) and PTB testing in the paediatric setting and were released in 2013 [16]. These updates were adopted by South Africa in 2014. Although these recommendations were applicable to cerebrospinal fluid (CSF), fine-needle aspirates (FNA), and lymph node and tissue biopsy specimen types, local testing on any purulent fluid was also included. This was based on findings where sensitivity varied for EPTB specimen types when stratified by the consistency of the specimen: 87% for thick/purulent versus 48% for clear/watery specimens [15].

Further improvements were introduced in 2017 with Cepheid’s launch of the Xpert^®^ MTB/RIF Ultra (Xpert Ultra) next-generation assay with improved sensitivity for smear-negative and HIV-associated TB [17]. The WHO recommendation was to use Xpert Ultra as the initial diagnostic test for all adults and children with signs and symptoms suggestive of TB [17]. Furthermore, Xpert Ultra was also recommended for the testing of selected EPTB specimen types [17,18]. In addition to enhancements in cartridge design, Ultra’s improved sensitivity is due to the detection of insertion sequences present in the multicopy, which generates a novel ‘*Mycobacterium tuberculosis* complex trace detected’ (Trace) call representing the lowest detectable organism load [17,18]. Based on available evidence, testing for PTB and EPTB across paediatric and adult settings were migrated to Xpert Ultra in South Africa from 2017, with the transition completed in the latter part of 2018 [7].

#### 2.3.2. Reporting of ‘*Mycobacterium tuberculosis* Complex Trace Detected’ Results

The WHO guidance on the management of Trace results suggested that for persons infected with HIV and those being investigated for EPTB and in the paediatric setting, Trace results should be considered as MTBC detected or true-positive results [17]. In alignment with the recommendations, at the implementation of Xpert Ultra, LIS interface rules were established to amend the instrument Trace results in order to report them as MTBC detected/rifampicin unsuccessful. This occurs for PTB specimens where the patient age is ≤5 years and for all EPTB specimens (irrespective of patient age). For all other scenarios (PTB specimens where the patient age is >5 years, the specimen origin is not defined, or the patient age is not known), the Trace results are reported as such.

#### 2.3.3. Specimen Collection Practice

Guided by a national testing algorithm, for individuals being investigated for presumptive TB, variation in specimen collection practice occurs: for diagnosis of PTB, a single sputum specimen is collected by all provinces, except for the Western Cape, where two specimens are collected upfront [18]. The second specimen is retained by the laboratories in the Western Cape; when the testing volume is insufficient or should further susceptibility testing or smear microscopy be required based on the initial testing results, the second specimen is immediately available for this. In provinces other than the Western Cape, further testing can only proceed once individuals have returned to the healthcare facility to produce the required additional specimen.

#### 2.3.4. Xpert^®^ MTB/RIF and Xpert^®^ MTB/RIF Ultra in the Context of the National Diagnostic Algorithm

TB diagnostic practice and treatment monitoring is guided by international and national clinical management algorithm [19,20]. In addition to Xpert (and later Xpert Ultra), other investigative diagnostic modalities, such as smear microscopy (utilised for monitoring the response to anti-TB therapy), molecular drug susceptibility testing for drugs other than rifampicin, mycobacterial culture, molecular tests to differentiate mycobacterial species, and phenotypic drug susceptibility testing, are used [20]. However, Xpert (Xpert Ultra) is the initial diagnostic test of choice when an individual is being investigated for suspected TB disease [18]. The Xpert (Xpert Ultra) result outcome then triggers follow-on clinical management steps and/or the conducting of additional investigations [20]. In the context of this analysis, only Xpert and Xpert Ultra data were included.

### 2.4. Data Preparation

The data extract was provided by the NHLS laboratory data repository and included the following variables: (i) reviewed date; (ii) age (in years); (iii) gender; (iv) province; (v) health district; (vi) instrument result; (vii) reported Xpert and Xpert Ultra MTBC result (MTBC Detected, MTBC Not Detected, MTBC Trace Detected, and Unsuccessful); (viii) RIF susceptibility result (RIF Resistance Detected, RIF Resistance Not Detected, and Unsuccessful); (ix) specimen category: PTB, EPTB or Unknown; (x) specimen type; (xi) specimen anatomical location; (xii) test assay utilised (Xpert or Xpert Ultra); and (xiii) numbers tested [19]. All rejected specimens and unverified results were excluded. The age and gender are provided by the healthcare worker on the laboratory request form and captured on the LIS when specimens are received for testing. The specimen type captured in the laboratory data was used to link the origin to the EPTB and consisted of the following: (i) aspirate/FNA; (ii) pus/abscess material; (iii) tissue; (iv) fluid; (v) urine; (vi) stool; and (vii) CSF. Sputum, gastric washing, and other pulmonary specimen types (bronchial brushings, bronchoalveolar lavage (BAL), nasopharyngeal and tracheal aspirate) were classified as PTB. Specimen types that were neither PTB nor EPTB were classified as unknown. This was achieved by developing a lookup table of all the unique specimen types captured in the data extract and pre-defining these as EPTB, PTB, or unknown. Patient age was categorised as follows: (i) 0–1, (ii) 2–3, (iii) 4–5, (iv) 6–7, (v) 8–10, (vi) 11–14, (vii) 15–25, (viii) 26–35, (ix) 36–45, (x) 46–55, (xi) 56–65, (xii) >65 years, and (xiii) unknown (where no age was provided or captured). The reviewed date of the result was used to extract the year and month.

The datasets were prepared and analysed using Microsoft Excel (Microsoft Corporation, Redmond, WA, USA), SAS 9.4 (SAS Institute, Cary, NC, USA) and Stata SE (Stata Corporation, College Station, TX, USA). The maps were created using ArcGIS (ESRI, Redlands, CA, USA).

### 2.5. Statistical Analysis

National and provincial tested volumes were reported per calendar year and month. For the MTBC detected specimens, the RIF resistance rate was determined per calendar year. In addition, the percentage of specimens that reported various MTBC result categories and RIF results was reported. The MTBC Detected, MTBC Trace Detected, and Unsuccessful rates were defined as follows: (i) $MTBC\;Detection\;Rate=\frac{MTBC\;Detected}{Number\;of\;Specimens\;Tested}$, (ii) $MTBC\;Trace\;Detected=\frac{MTBC\;Trace\;Detected}{Number\;of\;Specimens\;Tested}$, and (iii) $Unsuccessful\;test=\frac{Unsuccessful\;test}{Number\;of\;Specimens\;Tested}$. Similarly, the RIF resistance rate was defined as $RR=\frac{Resistance\;Detected}{MTBC\;Detected}$. The percentage change in the MTBC detection and RIF resistance rates between 2012 and 2022 was reported at the provincial level. The provincial population mid-year estimates were used to report the MTBC detection per 100,000 population ($\frac{MTBC\;Detected}{Provincial\;population}\times 100,000)$ [20]. Similarly, the RIF resistance rate per 100,000 population was reported ($\frac{RIF\;Resistance\;Detected}{Provincial\;population}\times 100,000)$ [20]. Tested volumes and MTBC detection rates were also segregated by Xpert and Xpert Ultra assays. Data were analysed by patient age and gender categories, by test assay (Xpert and Xpert Ultra), number of specimens tested, and MTBC detection and RIF resistance rates. The male-to-female ratio was calculated. After the transition to Xpert Ultra had been completed in 2018, the tested volumes and percentage of specimens with Trace results were reported by calendar year and by specimen type, and the analysis was repeated at the provincial level.

## 3. Results

### 3.1. National Testing Overview

Molecular testing for TB, performed with the GX instruments, is conducted across 173 laboratories (as of April 2023) within the NHLS network [10,11,21]. This number has varied between 165 and 173 laboratories based on operational decisions, such as availability of human resources, etc. [10,11,21]. As the GX instruments are available in varying sizes and module capacities (4-module platform, GXIV; 16 modules, GX16; 48 modules, GX48; and 80 modules, GX80), the instruments are placed either singly or in varying combinations to meet the very low-, low-, medium-, high- and very high-throughput needs, which vary by laboratory. The program currently comprises 232 (GXIV), 191 (GX16), 4 (GX48), and 8 (GX80) instruments, equating to a combined total of 4816 testing modules and resulting in a testing capacity exceeding 800,000 tests/month, factoring in laboratory operational hours and operating at ~80% efficiency. Figure 1 demonstrates the decentralisation of the GX testing laboratories across South Africa. To provide a contrast, the centralised mycobacteriology culture laboratories are also depicted.

During the study period, 23,740,668 specimens were tested. Of those, 1.9% failed to provide a result (unsuccessful test) and 9.4% detected MTBC (Table 1). Of the tests detecting MTBC, 5.9% detected rifampicin resistance (Table 2). The annual tested volumes ranged from 188,754 (2011) to 2,643,514 (2015) (Table 1). As testing was scaled up to reach national coverage by 2013, large year-on-year percentage increases were noted (237.1% in 2012 and a further 180.8% in 2013) (Table 1). The annual percentage change in tested volumes indicated a reduction of 8.6% and 9.1% from the previous year in 2016 and 2017, respectively. In 2020, a 22.4% percentage reduction in tested volumes was reported when compared to 2019 due to COVID-19. By 2022, a recovery of 2,534,050 specimens tested was recorded (which is like the values reported for 2015 and 2016) (Table 1).

The MTB detection rate was 15.9% in 2011 with the pilot commencement in the high-burden districts, decreasing annually and stabilising at ~9% between 2015 and 2020, then further declining to 7.3% by 2022 (Table 1).

With the program’s maturation, a declining trend in the rate of unsuccessful results was observed from 2.7% in 2013 to 1.2% by 2022 (Table 1).

From the end of 2017 to April 2023, there were 11,425,257 specimens tested using the Xpert Ultra assay (Appendix A). Of these, 191,480 Trace results (1.7%) were generated. Six provinces reported values above the national average. KwaZulu-Natal demonstrated the lowest Trace detection at 1.1%, while Western Cape had the highest, at 2.5% (Appendix A).

Over the decade, cyclical decreases in the tested volumes (and corresponding increases in the MTBC detection rate) were observed (Figure 2). These coincide yearly with the December/January and March/April months, respectively.

The national RIF resistance rate ranged from 7.3% in 2012 to 5.0% in 2021 (Table 2), which was reflective of a downward trend throughout the decade of testing. An increase in RIF unsuccessful rates occurred from 2018 with the transition to Xpert Ultra and the detection of Trace results, i.e., the detection of MTBC at the lowest detectable levels without the ability to report RIF resistance or susceptibility.

### 3.2. Provincial Overview

The national population estimate increased from 52.7 to 60.6 million between 2012 and 2022 [22,23]. Similarly, the number of specimens tested increased to 2.5 million in 2012 compared to 0.6 million in 2012 (Table 3). KwaZulu-Natal (the second most populous province) contributed 37.7% of the specimens tested by 2022; it was followed by Eastern Cape (11.0% of the population in 2022), at 15.6%; Gauteng (26.6% of the population in 2012), at 14.9%; and Western Cape (11.9% of the population in 2012), at 11.5% [22,23]. The remaining five provinces (31.5% of the population in 2022) contributed 20.3% of the tested specimens [22,23]. The trend remained unchanged from 2012 (Table 3). The MTBC detection (based on number of tested specimens) declined consistently across the provinces over the decade; it was the highest for KwaZulu-Natal and Mpumalanga (9.9%) and the lowest for the Western Cape (1.9%) and Free State (2.0%). In contrast, the MTBC detection (based on population estimates [22,23]) increased nationally from 171 per 100,000 (2012) to 306 per 100,000 (2022) (Table 3). By province in 2012, Free State, Northern Cape, and Eastern Cape reported detection rates of 412, 326, and 250 per 100,000 population, respectively. By 2022, the rates had increased to 656 (Eastern Cape), 638 (Northern Cape), and 580 (Western Cape) (Table 3). Eastern Cape reported the highest increase in detection per 100,000 population (from 2012), 406; it was followed by Western Cape with 380 and Northern Cape with 312. Five provinces demonstrated detection rates per 100,000 population in 2022 that were below the national average of 135 when compared to 2012. Free State was the only province demonstrating a decline in MTBC detection per 100,000 population over the decade and a decline in the number of specimens tested (Table 3).

Comparing 2012 and 2022, both the KwaZulu-Natal and the Mpumalanga provinces reported the highest percentage RIF resistance detection rates (Table 4). The percentage change in the RIF resistance rate between 2012 and 2022 ranged from a 0.3% (Western Cape) to a 4.4% (North West) decline. For 2012, the RIF resistance per 100,000 population ranged from 5 (Limpopo) to 26 (Free State), while for 2022 it ranged from 6 (Gauteng) to 35 (Eastern Cape). The overall RIF resistance per 100,000 population increased from 12 to 16 between 2012 and 2022, respectively (Table 4). The number of specimens reporting RIF resistance increased from 6456 in 2012 to 9735 by 2022 (a 1.4-fold increase). Eastern Cape reported the highest increase in RIF resistance detection per 100,000 population (18), followed by Western Cape (17) and Free State (16). Gauteng, KwaZulu-Natal, and Limpopo showed minimal changes in RIF resistance per 100,000 population from 2012, with Mpumalanga and North West demonstrating declines (Table 4).

### 3.3. Testing by Specimen Origin

PTB specimen processing declined year-on-year between 2016 and 2020. In contrast, EPTB specimen processing, included in the program since 2014, increased steadily and peaked at 111,533 in 2022 (Figure 3). Furthermore, COVID-19 impacted PTB specimen processing significantly more than EPTB testing in 2020 (23.3% versus 4.6% reduction) (Figure 3).

By anatomical site of collection, 92.5% of the specimens were pulmonary and 2.9% were extra-pulmonary in origin (Table 5). For 1,103,728 (4.6%) specimens, anatomical location could not be ascertained from the captured data. Overall, the detection rate for the EPTB specimens was 10.0%, compared to 9.4% for the PTB specimens (Table 5). The detection rates for the PTB specimens were higher using Xpert compared to Xpert Ultra (10.3% and 8.4%, respectively) while the detection rates for EPTB increased using Xpert Ultra compared to Xpert (10.2% and 9.5%, respectively (Table 5)).

Xpert Ultra testing generated ~191,480 Trace results. Of these, and based on the LIS interface rules, 21,776 PTB and EPTB specimens were subsequently reported as MTBC detected/RIF unsuccessful results. For 169,704 (1.5%) of the Trace results, these were reported as Trace.

For the specimens where origin was not known, the MTBC detection rate of 9.0% was reported.

#### 3.3.1. Pulmonary TB

Of the 21,951,794 PTB specimens, 98.9% were expectorated sputum (Table 6). The highest MTBC detection rate was reported for bronchoalveolar lavage (BAL) (11.4%), followed by bronchial brushings (9.6%) and sputum specimens (9.4%). Gastric aspirates reported the lowest MTBC detection rate (4.0%) (Table 6).

For specimens of PTB origin, bronchial brushings had the highest Trace detection rate (5.1%), followed by tracheal aspirate (2.5%) (Appendix A). Gastric aspirates reported a 1.7% Trace detection rate (Appendix A).

Rifampicin resistance varied by PTB specimen type and was the highest in BAL, at 8.1% (Table 6). The rates of unsuccessful testing were lowest for nasopharyngeal (0.3%) and BAL (0.7%) and highest for sputum specimens (1.8%) (Table 6).

#### 3.3.2. Extra-Pulmonary TB

Cerebrospinal fluid and fluids are the most common EPTB specimen types, constituting 53.9% and 32.1%, respectively, of the 685,146 specimens tested (Table 7). The MTBC detection rate, when analysed by EPTB specimen type, ranged from 3.5% (CSF) to 35.8% (aspirate/FNA) (Table 7). An MTBC detection rate above 20% was reported for pus/abscess material (28.4%). Lower MTB detection rates were reported for the remaining specimen types, ranging from 7.9% (stool) to 16.8% (tissue) (Table 7). Pus/abscess material had the highest unsuccessful rates (2.0%), with tissue reporting the lowest at 0.9% (Table 7).

For the specimens of EPTB origin, fluid, aspirate/FNA, and tissue reported Trace results of 4.7%, 4.3%, and 3.3%, respectively. For CSF specimens, the Trace results contributed 1.7%. The lowest Trace was reported for pus/abscess material, at 1.2%. For the specimens of unknown origin, the Trace results contributed 1.6% (Appendix A).

Sixty-nine percent of the processed CSF specimens were collected from those aged between 15 and 55, followed by 10.8% from those aged 0–1 year (Appendix A). The detection of MTBC peaked at 4.7% for those aged 8–10 years, with lowest yield reported for the >65 years category, at 1.5%. In the 0–1 year category, the detection rate was 2.2% (Appendix A).

For the EPTB specimens, the MTBC detection rate also varied by anatomical site. For abdominal EPTB, aspirates/FNA yielded marginally higher detection rates than the other specimen types (Table 8). The analysis of specimens derived from lymph nodes (aspirate/FNA, pus/abscess material and tissue) yielded MTBC detection rates > 40%, except for the fluid collections. Pericardial fluid and tissue were positive in 33–38% of specimens. Pleural space fluid or aspirates yielded similar detection rates (~21%) and doubled if pleural specimens were purulent in nature (43.2%). The detection from pleural tissue was low (6.5%). Skeletal aspirates were more likely to detect MTBC. Where anatomical location was not captured, both aspirates/FNA and pus/abscess specimen types demonstrated detection rates > 28% (Table 8). Tissue specimens derived from lymph nodes had significantly higher positivity rates (42.9%) compared to the other anatomical origins (Table 8).

### 3.4. Demographic Analysis

During the decade of testing, 71.0% of specimens tested were from individuals in the age categories of 15–25 (15.0%), 26–35 (22.4%), 36–45 (19.2%), and 46–55 years (14.4%) (Appendix A). The MTBC detection rate ranged from 1.3% (6–7 years) to 12.4% (26–35 years). Specimens submitted from the paediatric setting had the highest MTBC detection rates in the 0–1 year group. For specimens where age was not known, a detection rate of 10.1% was reported. RIF resistance varied between 3.8% (>65 years) and 8.9% (0–1 year group).

Gender was recorded for 93.3% of tested specimens. Of those, a male-to-female ratio of 0.85:1 was reported (Appendix A). A higher MTBC detection rate (12.0% versus 6.7%) was reported for males than females, respectively. In contrast, a higher RIF resistance rate was reported for females than males (6.5% versus 5.4%). In the tested specimens lacking a gender demographic, 13.0% reported MTBC detection and 6.4% of these had RIF resistance.

## 4. Discussion

The objectives of this study were to describe the scale-up of molecular TB diagnostics in South Africa and to illustrate the changes over a decade of testing. The results show the changes over time, by specimen type, age, and gender, and they highlight the utility of a single laboratory data repository linked to an LIS connecting all the public sector testing laboratories and testing platforms servicing the integrated pathology diagnostic needs of the state sector population of South Africa.

Since the introduction of the Xpert assay, over 23 million tests have been conducted. Significant year-on-year volume increases were noted during the national scale-up of Xpert testing. Cyclical annual trends of decreases in testing, coupled with increases in the detection rate of the specimens that are tested, are noted during festive and holiday periods, suggesting only the very ill are tested. These annual trends are also reflected in previous studies [24,25].

The introduction of testing for EPTB specimen types and the broadening of testing to include paediatric specimens also contributed to the tested volume increases. The trend of a reduction in the number of tested specimens may be attributed to the decreasing incidence of TB seen in South Africa from 2011 (892 per 100,000 population) to 2021 (513 per 100,000 population) [1,26,27] but may also be attributed to a lack of testing being performed, particularly amongst men living with HIV. The 2018 South African prevalence survey found high levels of TB in this group [2,28]. Our results show that males receive less testing than females yet have rates of MTBC detection that are almost twice as high, and these results are consistent with those of other studies [29]. People aged 15–55 also showed high rates of MTBC detection; this too is reflected in previous studies and may be due to these individuals being less able to access health services as they are at work while the services are operational [30].

Following the WHO declaration defining COVID-19 as a global pandemic in March 2020 [31], the first confirmed case was diagnosed in South Africa on 5 March [32], with lockdown measures imposed by the government 18 days later to control viral transmission [33]. Lockdown strategies, social distancing rules, and community containment measures for COVID-19 reported negative impacts on the diagnoses and treatment of communicable diseases [34], with a significant impact on TB molecular tested numbers in 2020 and 2021. Such measures not only restricted mobility and access to healthcare facilities but also caused staffing constraints which compromised the provision of services [35]. Local challenges were shared globally. The COVID-19 pandemic had a damaging impact on access to TB diagnosis and treatment, affecting the burden of disease globally with the number of newly diagnosed people with TB decreasing to 5.8 million by 2020 [1]. However, the results show that the national TB testing program has recovered from the impact of the COVID-19 epidemic on TB testing levels. Although they make up a small proportion of specimens collected and tested, the collection of extra-pulmonary specimens did not sustain the decreases shown with PTB during this period. Specimens to diagnose EPTB require invasive sampling, which is performed in the higher tiers of healthcare (hospitals). We also see increases in MTBC detection rates during this time, suggesting that only the sickest individuals were seeking care.

Trace results were first reported in 2017, with the implementation of Xpert Ultra. As Xpert (and Xpert Ultra) are not able to distinguish the viability of the detected organism [36], the improved assay sensitivity may result in the detection of non-viable bacilli from a previous TB episode [36]. Thus, Trace results are hard to interpret clinically [37]. The 2018 South African TB prevalence survey found high rates of individuals with recurrent TB, and furthermore, an individual may not know if they had a previous TB episode [2]. As such, further clinical evaluation, investigations, and knowing the patient context (including prior treatment history and probability of relapse) are still necessary to understand the interpretation of such results.

Although RIF resistance detection rates (as a percentage of MTBC detected specimens) declined nationally over the study period, the rate per 100,000 population increased to 16 from 12 in 2012. A prospective cohort study performed in three South African provinces during our study period showed only 8% of patients completed the entire RIF-resistant TB care cascade as intended [38], which also suggested that this decrease may not be reflective of an actual decrease in RIF-resistant TB prevalence. The South African Tuberculosis Drug-Resistant TB survey from 2012 to 2014 reported an increase in RIF resistance prevalence compared to the previous survey conducted in 2011–2002 [28]. The increase was predominantly seen among new TB cases. Our analysis could not differentiate the proportion of RIF resistance in the newly diagnosed compared to those with previous TB episodes or untreated cases as this information was either not always provided by the healthcare worker or not captured within the LIS.

Sputum is by far the most common PTB specimen type collected, although it also has the highest rate of unsuccessful tests. This could be due in part to small sample sizes for other PTB specimen types. That said, it also has one of the highest MTBC detection rates. Collecting other PTB specimen types (like BAL, bronchial brushing, gastric aspirate, and tracheal aspirate) is time- and resource-intensive [39]. Although we tested small numbers of BAL specimens, their MTBC detection rate is high, and it has previously been shown to be a useful diagnostic tool for PTB [40]. Gastric aspirates saw the lowest rate of MTBC detection, and a small number of specimens were collected and tested. Gastric aspirates were found to have low sensitivity (but high specificity) in a meta-analysis [40,41]. Its sensitivity and specificity are even lower in children under 4 years of age.

Of all the EPTB specimen types, CSF is the most common collected type, but it reports the lowest MTBC detection rate. As CSF is a critical specimen limited by collection volume (especially in paediatric cases) and numerous investigations are requested as part of the differential diagnosis, volume limits may compromise the sensitivity of detection [42]. Despite its low sensitivity, Xpert (and Xpert Ultra) is still the WHO’s recommended initial diagnostic test to diagnose TB meningitis [43]. Aspirate/FNA had the highest MTBC detection rate; this specimen type has been shown to have considerable specificity and sensitivity with the Xpert (Xpert Ultra) assay [43]. Fluid has been shown to have a high specificity but relatively low sensitivity with the Xpert (Xpert Ultra) assay [44]. We report fluid having a moderate MTBC detection rate across all anatomical sites, except if it is of skeletal origin. The reported sensitivity rates for pleural fluid are particularly low [45], and the WHO recommendations for EPTB testing with Xpert and Xpert Ultra do not include pleural fluid. However, we report a detection rate of over 20% for pleural fluid testing, and the detection rate doubles if the pleural fluid is purulent in nature. Stool and urine specimen types have not been implemented in the program, but requests are received, linked to research-based activities. As such, both show low counts of tested specimens.

By specimen type, more invasively collected PTB specimens, such as bronchial brushings and tracheal aspirates, demonstrated higher Trace detection values. For EPTB specimen types, fluids and aspirates/FNA, followed by tissue, demonstrated a doubling of the Trace detection compared to the overall rate. For fluid and purulent material, an inverse relationship exists between MTBC detection and the Trace rates, i.e., pus/abscess material has a high MTBC detection rate but low Trace detection and vice versa for fluid, suggesting that the bacillary load is higher in purulent material.

Despite the successes of the national GX program, gaps remain in the service offering: (1) Provision of testing remains confined to laboratories, and further decentralisation in the context of the true point of care should be explored, aiming for improvements in the linkage to care. (2) Despite its footprint and multi-disease testing potential, GX use is largely confined to TB molecular testing without full integration with HIV testing and monitoring (currently conducted on separate platforms). (3) There is a need to introduce stool (for the diagnosis of paediatric PTB) and urine specimens as additional specimen testing types. (4) Interventions aimed at improving the quality of collected specimens with respect to adequacy and volume are needed to ensure that assay sensitivity is not compromised. (5) Sputum collection practices need to be aligned across provinces to collect two upfront sputum specimens and mitigate losses in the TB care cascade [46,47,48].

A great strength of this study is the large amount of data available. The more than 23 million tests allow the stratification by geographic location, specimen type, age, and gender to gain insight into South Africa’s national TB testing program and the rollout of the Xpert (Xpert Ultra) testing platforms. Limitations include lack of clinical outcomes or specific details, such as previous TB episodes or anti-TB therapy; as such, it is difficult to draw clinical conclusions. An additional limitation is that these data are reliant on what is (or is not) captured by the LIS. We see higher than ideal rates of missing demographic and specimen detail data. These data also lack a unique patient identifier, such that individuals may be represented in the data multiple times, and we cannot relate testing rates to the TB burden. The analysis focused only on Xpert (Xpert Ultra) data and did not include correlations with other diagnostic modalities within the context of the national algorithm (such as mycobacterial culture and differentiation of mycobacterial species) where these were available. Finally, these data consist only of individuals who seek care and have a positive symptom screen showing that they are eligible to receive specimen testing. Recent research has shown there may exist a phase referred to as subclinical TB, in which individuals are asymptomatic but still able to transmit TB [49]. As such, in March 2023, the NDOH introduced a national TB Recovery Plan, which included a novel intervention of targeted universal testing (TUTT) in high-risk groups [50]. This is a change from the previous strategy of only offering a TB test to those with symptoms that include cough, chest pain, fatigue, weight loss, and night sweats [51]. With TUTT, everyone at a high risk of contracting TB receives a test [51]. A local study reported that TUTT identified more TB patients than the standard of care (20.7 patients with TB per clinic month versus 18.8, respectively) [52]. The NDOH reported a 17% increase in TB diagnosis in clinics with TUTT compared to the existing approach [50,51]. TUTT will result in patients being offered TB tests even if asymptomatic [50,51].

## 5. Conclusions

The analysis of the molecular TB testing data, which was implemented over a decade ago, has highlighted testing gaps in males, low detection rates in paediatric specimens, and the likely effects of specimen quality and adequacy on detection yield. Access to a national, standardised data repository which encompasses testing for the majority of South Africa’s population helps to provide information on access to services, interventions, the need to refine algorithms and practices, and the operational aspects required for programmatic improvements.

## Figures and Tables

**Figure 1 diagnostics-13-03253-f001:**
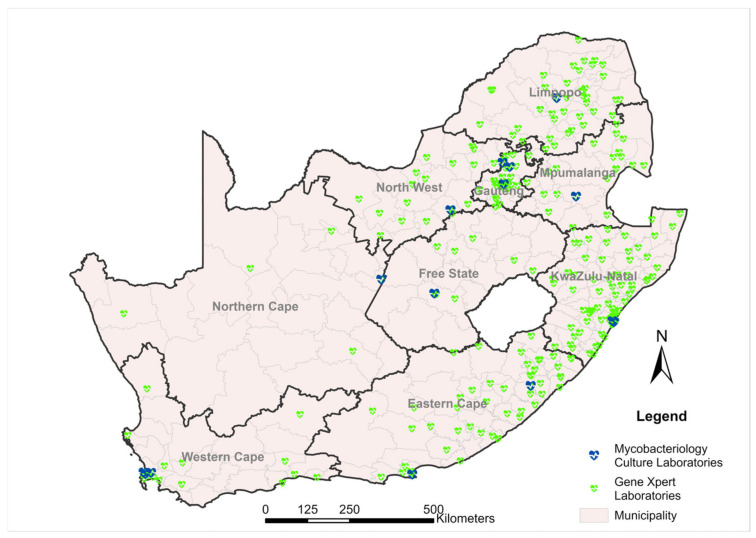
Decentralised footprint of 173 GeneXpert testing laboratories supporting molecular tuberculosis testing by the National Health Laboratory Service, South Africa. The laboratories that offer mycobacteriology culture testing are also depicted. The number of testing laboratories varies by province based on the respective provincial population.

**Figure 2 diagnostics-13-03253-f002:**
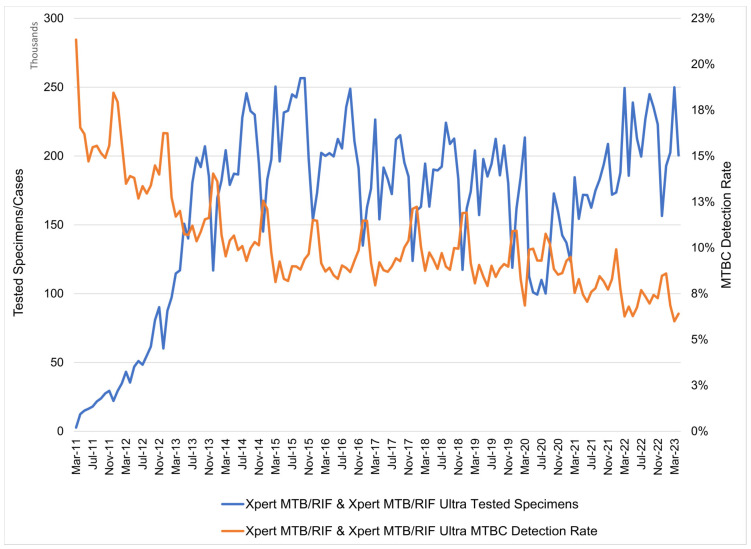
GeneXpert testing in South Africa with number of specimens and *Mycobacterium tuberculosis* complex detection rate (indicated on the secondary y-axis), by month. Transition from Xpert MTB/RIF to Xpert MTB/RIF Ultra was phased commencing October 2017 and fully completed by April 2018. MTBC: *Mycobacterium tuberculosis* complex.

**Figure 3 diagnostics-13-03253-f003:**
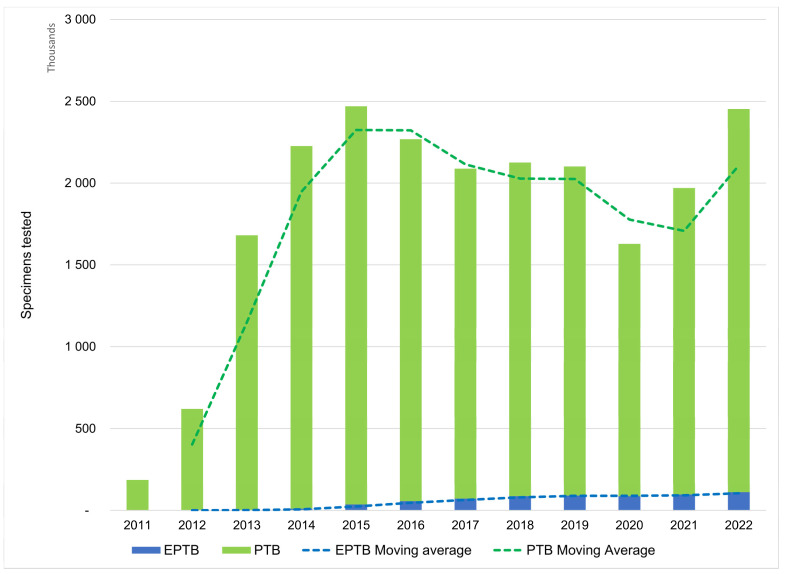
Number of specimens tested on GeneXpert platforms by anatomical origin (pulmonary and extra-pulmonary) by calendar year. Specimens that could not be categorised as either of pulmonary or extra-pulmonary origin (*n* = 1,103,728) were excluded. COVID-19 lockdown and restrictive containment measures were introduced in March 2020. EPTB: extra-pulmonary tuberculosis; PTB: pulmonary tuberculosis.

**Table 1 diagnostics-13-03253-t001:** Number of specimens processed annually on the GeneXpert platforms by result category and as unsuccessful tests.

Year	Specimens Tested *n* = (%)	% Annual Change from Previous Year ^&^	MTBC Detected *n* = (%)	MTBC Not Detected *n* = (%)	Unsuccessful Test
2011	188,754 (0.8)		29,947 (15.9)	155,904 (82.6)	2903 (1.5)
2012	636,241 (2.7)	237.1%	90,279 (14.2)	535,694 (84.2)	10,268 (1.6)
2013	1,786,862 (7.5)	180.8%	208,645 (11.7)	1,530,727 (85.7)	47,490 (2.7)
2014	2,384,710 (10.0)	33.5%	249,708 (10.5)	2,075,309 (87.0)	59,693 (2.5)
2015	2,643,514 (11.1)	10.9%	245,393 (9.3)	2,341,811 (88.6)	56,310 (2.1)
2016	2,416,517 (10.2)	−8.6%	223,454 (9.2)	2,144,461 (88.7)	48,602 (2.0)
2017	2,197,555 (9.3)	−9.1%	208,924 (9.5)	1,953,514 (88.9)	33,475 (1.5)
2018	2,199,299 (9.3)	0.1%	214,044 (9.7)	1,910,477 (86.9)	37,711 (1.7)
2019	2,179,502 (9.2)	−0.9%	197,138 (9.0)	1,901,765 (87.3)	43,753 (2.0)
2020	1,690,520 (7.1)	−22.4%	153,791 (9.1)	1,478,496 (87.5)	32,854 (1.9)
2021	2,037,432 (8.6)	20.5%	163,641 (8.0)	1,820,550 (89.4)	26,987 (1.3)
2022	2,534,050 (10.7)	24.4%	185,252 (7.3)	2,285,341 (90.2)	31,322 (1.2)
2023	845,712 (3.6)		58,261 (6.9)	768,947 (90.9)	8123 (1.0)
Total	23,740,668 (100.0)		2,228,477 (9.4)	20,902,996 (88.0)	439,491 (1.9)

MTBC: *Mycobacterium tuberculosis* complex. ^&^ Data only reported up to April 2023; hence, the percentage change is not included for 2023.

**Table 2 diagnostics-13-03253-t002:** Number of specimens processed annually on the GeneXpert platforms reporting detection of *Mycobacterium tuberculosis* complex stratified by rifampicin resistance.

Year	MTBC Detected *n* = (%)	RIF Resistance Detected *n* = (%)	RIF Resistance Not Detected *n* = (%)	RIF Unsuccessful *n* = (%)
2011	29,947 (15.9)	2124 (7.1)	27,422 (91.6)	293 (1.0)
2012	90,279 (14.2)	6546 (7.3)	81,810 (90.6)	1285 (1.4)
2013	208,645 (11.7)	13,802 (6.6)	188,383 (90.3)	5380 (2.6)
2014	249,708 (10.5)	16,316 (6.5)	226,941 (90.9)	6101 (2.4)
2015	245,393 (9.3)	14,999 (6.1)	226,898 (92.5)	3296 (1.3)
2016	223,454 (9.2)	13,815 (6.2)	207,105 (92.7)	2395 (1.1)
2017	208,924 (9.5)	12,183 (5.8)	194,384 (93.0)	2278 (1.1)
2018	214,044 (9.7)	11,375 (5.3)	198,321 (92.7)	4282 (2.0)
2019	197,138 (9.0)	10,431 (5.3)	180,550 (91.6)	6132 (3.1)
2020	153,791 (9.1)	8129 (5.3)	140,636 (91.4)	5017 (3.3)
2021	163,641 (8.0)	8169 (5.0)	150,773 (92.1)	4677 (2.9)
2022	185,252 (7.3)	9375 (5.1)	170,363 (92.0)	5502 (3.0)
2023	58,261 (6.9)	3189 (5.5)	53,257 (91.4)	1814 (3.1)
Total	2,228,477 (9.4) ^#^	130,453 (5.9)	2,046,843 (91.8)	48,452 (2.2)

MTBC: *Mycobacterium tuberculosis* complex; RIF Rifampicin. ^#^ Includes 2729 specimens where no rifampicin result was generated for various reasons.

**Table 3 diagnostics-13-03253-t003:** Analysis of number of tested specimens by province stratified by *Mycobacterium tuberculosis* complex detection rate expressed as a percentage and rate per 100,000 population (change between 2012 and 2022).

Province	2012				2022				Change in MTBC Detected Rate (%)	Change in Population (%)	Change in MTBC Detected per 100,000 Population
	Specimens Tested *n* = (%)	MTBC Detected *n* = (%)	Population *n* = (%)	MTBC Detected per 100,000 Population	Specimens Tested *n* = (%)	MTBC Detected *n* = (%)	Population *n* = (%)	MTBC Detected per 100,000 Population			
EC	104,574 (16.4)	16,501 (15.8)	6,594,537 (12.5)	250	396,001 (15.6)	43,778 (11.1)	6,676,691 (11.0)	656	−4.7	1.2	406
FS	88,222 (13.9)	11,501 (13.0)	2,793,604 (5.3)	412	73,544 (2.9)	8119 (11.0)	2,921,611 (4.8)	278	−2.0	4.6	−134
GP	84,200 (13.2)	11,137 (13.2)	12,630,422 (24.0)	88	377,698 (14.9)	22,620 (6.0)	16,098,571 (26.6)	141	−7.2	27.5	53
KZN	160,732 (25.3)	22,032 (13.7)	10,406,665 (19.7)	212	955,702 (37.7)	35,870 (3.8)	11,538,325 (19.0)	311	−9.9	10.9	99
LP	35,417 (5.6)	3937 (11.1)	5,447,963 (10.3)	72	119,198 (4.7)	7074 (5.9)	5,941,439 (9.8)	119	−5.2	9.1	47
MP	26,874 (4.2)	4003 (14.9)	4,114,293 (7.8)	97	146,530 (5.8)	7306 (5.0)	4,720,497 (7.8)	155	−9.9	14.7	58
NW	37,456 (5.9)	5398 (14.4)	3,574,090 (6.8)	151	101,860 (4.0)	10,324 (10.1)	4,186,984 (6.9)	247	−4.3	17.1	96
NC	25,292 (4.0)	3800 (15.0)	1,164,483 (2.2)	326	73,137 (2.9)	8345 (11.4)	1,308,734 (2.2)	638	−3.6	12.4	312
WC	73,474 (11.5)	11,970 (16.3)	5,973,197 (11.3)	200	290,380 (11.5)	41,816 (14.4)	7,212,142 (11.9)	580	−1.9	20.7	380
Total	636,241 (100.0)	90,279 (14.2)	52,699,253 (100.0)	171	2,534,050 (100.0)	185,252 (7.3)	60,604,992 (100.0)	306	−6.9	15.0	135

MTBC: *Mycobacterium tuberculosis* complex; EC: Eastern Cape; FS: Free State; GP: Gauteng Province; KZN: KwaZulu-Natal; LP: Limpopo; MP: Mpumalanga; NW: North West; NC: Northern Cape; WC: Western Cape.

**Table 4 diagnostics-13-03253-t004:** Analysis of number of *Mycobacterium tuberculosis* complex detected specimens by province stratified by rifampicin resistance rate expressed as a percentage and rate per 100,000 population (change between 2012 and 2022).

Province	2012				2022				Change in RIF Resistance Detected Rate (%)	Change in Population (%)	Change in RIF Resistance per 100,000 Population
	MTBC Detected *n* = (%)	RIF Resistance Detected *n* = (%)	Population *n* = (%)	RIF Resistance per 100,000 Population	MTBC Detected *n* = (%)	RIF Resistance Detected *n* = (%)	Population *n* = (%)	RIF Resistance per 100,000 Population			
EC	16,501 (18.3)	1118 (6.8)	6,594,537 (12.5)	17	43,778 (23.6)	2311 (5.3)	6,676,691 (11.0)	35	−1.5%	1.2	18
FS	11,501 (12.7)	734 (6.4)	2,793,604 (5.3)	26	8119 (4.4)	303 (3.7)	2,921,611 (4.8)	10	−2.7%	4.6	16
GP	11,137 (12.3)	773 (6.9)	12,630,422 (24.0)	6	22,620 (12.2)	1024 (4.5)	16,098,571 (26.6)	6	−2.4%	27.5	0
KZN	22,032 (24.4)	2040 (9.3)	10,406,665 (19.7)	20	35,870 (19.4)	2349 (6.5)	11,538,325 (19.0)	20	−2.7%	10.9	0
LP	3937 (4.4)	260 (6.6)	5,447,963 (10.3)	5	7074 (3.8)	331 (4.7)	5,941,439 (9.8)	6	−1.9%	9.1	1
MP	4003 (4.4)	397 (9.9)	4,114,293 (7.8)	10	7306 (3.9)	403 (5.5)	4,720,497 (7.8)	9	−4.4%	14.7	−1
NW	5398 (6.0)	396 (7.3)	3,574,090 (6.8)	11	10,324 (5.6)	376 (3.6)	4,186,984 (6.9)	9	−3.7%	17.1	−2
NC	3800 (4.2)	240 (6.3)	1,164,483 (2.2)	21	8345 (4.5)	346 (4.1)	1,308,734 (2.2)	26	−2.2%	12.4	5
WC	11,970 (13.3)	588 (4.9)	5,973,197 (11.3)	10	41,816 (22.6)	1932 (4.6)	7,212,142 (11.9)	27	−0.3%	20.7	17
Total	90,279 (100.0)	6546 (7.3)	52,699,253 (100.0)	12	185,252 (100.0)	9375 (5.1)	60,604,992 (100.0)	16	−2.2%	15.0	4

MTBC: *Mycobacterium tuberculosis* complex; RIF: Rifampicin; EC: Eastern Cape; FS: Free State; GP: Gauteng Province; KZN: KwaZulu-Natal; LP: Limpopo; MP: Mpumalanga; NW: North West; NC: Northern Cape; WC: Western Cape.

**Table 5 diagnostics-13-03253-t005:** Number of specimens tested by Xpert MTB/RIF and Xpert MTB/RIF Ultra assays by anatomical origin (pulmonary, extra-pulmonary, or origin unknown) and respective *Mycobacterium tuberculosis* complex detection rates.

Assay	PTB Specimens Tested *n* = (%)	EPTB Specimens Tested *n* = (%)	Specimens of Unknown Origin Tested *n* = (%)	Total	MTBC Detected for PTB Specimens *n* = (%)	MTBC Detected for EPTB Specimens *n* = (%)	MTBC Detected for Specimens of Unknown Origin *n* = (%)
Xpert^®^ MTB/RIF (from April 2011 to 2017/2018 transition)	11,423,713 (92.8)	179,345 (1.5)	712,353 (5.8)	12,315,411 (51.9)	1,178,841 (10.3)	17,122 (9.5)	66,342 (9.3)
Xpert^®^ MTB/RIF Ultra (from 2017/2018 transition to April 2023)	10,528,081 (92.1)	505,801 (4.4)	391,375 (3.4)	11,425,257 (48.1)	881,333 (8,4)	51,388 (10.2)	33,451 (8.5)
Total	21,951,794 (92.5)	685,146 (2.9)	1,103,728 (4.6)	23,740,668 (100)	2,060,174 (9.4)	68,510 (10.0)	99,793 (9.0)

MTBC: *Mycobacterium tuberculosis* complex; PTB: pulmonary tuberculosis; EPTB: extra-pulmonary tuberculosis.

**Table 6 diagnostics-13-03253-t006:** Analysis of pulmonary tuberculosis specimens tested, *Mycobacterium tuberculosis* complex detection, rifampicin resistance, and unsuccessful rates by specimen type.

PTB Specimen Type	Tested Specimens *n* = (%)	MTBC Detected *n* = (%)	MTBC Not Detected *n* = (%)	RIF Resistance Detected *n* = (%)	Unsuccessful Test *n* = (%)
Bronchial brushings	872 (0.0)	84 (9.6)	769 (88.2)	5 (6.0)	11 (1.3)
BAL	9924 (0.0)	1133 (11.4)	8507 (85.7)	92 (8.1)	73 (0.7)
Gastric aspirate	191,198 (0.9)	7702 (4.0)	179,653 (94.0)	486 (6.3)	3123 (1.6)
Nasopharyngeal	1166 (0.0)	57 (4.9)	1101 (94.4)	3 (5.3)	3 (0.3)
Sputum	21,714,452 (98.9)	2,048,812 (9.4)	19,104,211 (88.0)	119,273 (5.8)	399,638 (1.8)
Tracheal aspirate	34,182 (0.2)	2386 (7.0)	30,971 (90.6)	130 (5.4)	388 (1.1)
Total	21,951,794 (100.0)	2,060,174 (9.4)	19,325,212 (88.0)	119,989 (5.8)	403,236 (1.8)

BAL: Bronchoalveolar lavage; MTBC: *Mycobacterium tuberculos* is complex; PTB: pulmonary tuberculosis; RIF: rifampicin.

**Table 7 diagnostics-13-03253-t007:** Analysis of extrapulmonary tuberculosis specimens tested, *Mycobacterium tuberculosis* complex detection, rifampicin resistance, and unsuccessful rates by specimen type.

EPTB Specimen Type	Tested Specimens *n* = (%)	MTBC Detected *n* = (%)	MTBC Not Detected *n* = (%)	RIF Resistance Detected *n* = (%)	Unsuccessful Test *n* = (%)
Aspirate/FNA	15,918 (2.3)	5700 (35.8)	9933 (62.4)	447 (7.8)	239 (1.5)
CSF	369,251 (53.9)	12,869 (3.5)	351,014 (95.1)	832 (6.5)	5088 (1.4)
Fluid	220,118 (32.1)	32,398 (14.7)	183,796 (83.5)	1656 (5.1)	3462 (1.6)
Pus/Abscess	41,909 (6.1)	11,910 (28.4)	29,122 (69.5)	1213 (10.2)	844 (2.0)
Stool ^#^	458 (0.1)	36 (7.9)	415 (90.6)	3 (8.3)	7 (1.5)
Tissue	24,014 (3.5)	4044 (16.8)	19,722 (82.1)	243 (6.0)	209 (0.9)
Urine ^#^	13,478 (2.0)	1553 (11.5)	11,589 (86.0)	75 (4.8)	332 (2.5)
Total	685,146 (100.0)	68,510 (10.0)	605,591 (88.4)	4469 (6.5)	10,181 (1.5)

FNA: fine needle aspirate; CSF: cerebrospinal fluid; MTBC: *Mycobacterium tuberculosis* complex; EPTB: extra-pulmonary tuberculosis; RIF: rifampicin. ^#^ Tested under research-based studies. Both stool and urine were not officially adopted specimen types within the program.

**Table 8 diagnostics-13-03253-t008:** *Mycobacterium tuberculosis* complex detection rate by specimen type and anatomical site affected in extra-pulmonary TB.

EPTB Specimen Type	MTBC Detection Rate by Anatomical Site in Extra-Pulmonary TB *n* = Specimens Tested (Detection Rate %)					
	Abdominal ^#^	Lymph Nodes ^@^	Pericardial	Pleural	Skeletal ^^^	Unknown
Aspirate/FNA	165 (23.6)	1377 (44.1)		32 (21.9)	30 (50.0)	14,314 (35.2)
Fluid	2029 (11.4)	118 (11.9)	182 (37.9)	6320 (21.7)	2792 (4.4)	208,677 (14.7)
Pus/Abscess	57 (17.5)	367 (41.4)	1 (0.0)	37 (43.2)	189 (36.0)	41,258 (28.3)
Tissue	116 (19.8)	189 (42.9)	3 (33.3)	46 (6.5)	1153 (17.3)	22,507 (16.6)

^#^ Abdominal includes ascitic, peritoneal, intra-abdominal, gynaecological, and urogenital specimens. ^@^ Lymph nodes include axillary, cervical, neck, head, clavicular, submandibular, sublingual, submental, chest, breast, inguinal, and popliteal. ^^^ Skeletal includes bone, joint, skin, jaw, shoulder, elbow, wrist, hip, knee, psoas, back, limbs, ankle, and buttock.

## Data Availability

The authors do not have permission to share the data.

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
