# Peer review of "More Than a Decade of GeneXpert® Mycobacterium tuberculosis/Rifampicin (Ultra) Testing in South Africa: Laboratory Insights from Twenty-Three Million Tests"

_diagnostics, 2023, doi:10.3390/diagnostics13203253_

Round 1
Reviewer 1 Report
In their manuscript, Silva et al. conduct a retrospective analysis spanning over a decade, examining South Africa's shift to molecular TB diagnostics via the GeneXpert (GX) platform. The study illuminates diverse testing patterns and detection rates associated with different specimen origins and patient demographics. The manuscript boasts a robust structure and commendable writing quality, presenting intriguing insights such as the pronounced gender disparities in MTBC detection rates. I believe that this research will resonate with the clinical TB research community and aligns seamlessly with the journal Diagnostics. I'd like to offer a few minor suggestions for consideration:
Minor comments:
Ln 288-296: The use of "age" here can be ambiguous. Are we discussing patient age or the specimen's storage duration? I'd recommend providing a clearer definition. Additionally, in Ln 288, it might be beneficial to specify that detecting MTBC at its most minimal is referred to as a "trace" of MTB positivity, as indicated in Table S2.
Ln 34: There appears to be an extraneous indent.
Ln 232: To enhance clarity, consider revising to "RIF resistance/susceptibility.
Reviewer 2 Report
The authors describe the rollout and current state of South Africa’s GeneXpert molecular diagnostic program for tuberculosis with an analysis of 23 million tests. There is an in-depth analysis that can help better programmatic management of TB.
Comment 1: The authors have not mentioned the smear results for these patients, as under national TB program settings smears are usually done along with Xpert testing. This can help identify the proportion of MTB detected (also RIF resistance) among smear-negative cases. Also, suspicion of non-tuberculous mycobacteria (NTM) for smear-positive, MTB not detected on Xpert is only possible if smear results are available. If smears for AFB were not done or results are not available, it should be listed as a limitation in the discussion
WHO updated guidelines recommend molecular diagnostics as initial tests for diagnosis of TB and drug resistance, however according to the WHO “Nonetheless, conventional microscopy and culture remain necessary to monitor a patient’s response to treatment. Culture is still important in the diagnosis of paediatric and extrapulmonary TB from paucibacillary samples, and in the differential diagnosis of non-tuberculous mycobacteria (NTM) infection” (WHO operational Handbook on tuberculosis, Rapid diagnostics for tuberculosis detection, 2021 update).
Comment 2: The history of previous TB treatment is also very important to properly define the proportion of RIF resistance cases. It is not clear what is the exact proportion of RIF resistance in newly diagnosed and previously untreated cases. Again if this information is not available, it should be mentioned in the discussion
Reviewer 3 Report
Congratulations for the major undertaking of analyzing the dataset on over 23 million XpertMTB/RIF tests carried out in South Africa in the past decade or so. This manuscript is well prepared and I have only the following suggestions for minor revisions:
* Ethical considerations are not mentioned - please include information about the approval(s).
* Line 237/provincial overview: how would these data look like if they were expressed as a rate of tests per 100,000 population for each province for selected years (e.g., 2012 and 2022, as presented in Table 3)? In addition to the proportions of provincial 'contribution' to the testing work load, this rate could reveal estimates of presumptive TB rate per 100,000 population in a province (or rate of specimens collected and reaching lab per 100,000 population) in a given year and the time trends of these rates. These data would be helpful when analyzing the test positivity rates.
* There is no mention of the financial considerations of carrying out all these tests (cartridges, labor, lab infrastructure, etc.). An initial, pre-roll out of molecular testing, cost-effectiveness study is referenced - thank you. Could this be a topic for another paper to assess whether the conclusion of 'early days' did stand the test of times?
* It would also be most interesting and informative for laboratory and TB staff globally to hear the South African lessons learned regarding the transition from (sputum) microscopy to automated molecular testing? What implications to staffing levels, lab safety (any data on incidence of TB infection and TB disease and TST conversions among laboratory staff?), turn-around-times, sample transport systems, lab and other HCW satisfaction, etc,. has this transition had?
